# Differences in Demographic and Radiographic Characteristics between Patients with Visible and Invisible T1 Slopes on Lateral Cervical Radiographic Images

**DOI:** 10.3390/jcm11020411

**Published:** 2022-01-14

**Authors:** Sadayuki Ito, Hiroaki Nakashima, Akiyuki Matsumoto, Kei Ando, Masaaki Machino, Naoki Segi, Hiroyuki Tomita, Hiroyuki Koshimizu, Shiro Imagama

**Affiliations:** 1Department of Orthopedic Surgery, Nagoya University Graduate School of Medicine, Nagoya 466-8560, Japan; sadaito@med.nagoya-u.ac.jp (S.I.); andokei@med.nagoya-u.ac.jp (K.A.); masaaki_machino_5445_2@yahoo.co.jp (M.M.); naoki.s.n@gmail.com (N.S.); hiro_tomi_1031@yahoo.co.jp (H.T.); love_derika@yahoo.co.jp (H.K.); imagama@med.nagoya-u.ac.jp (S.I.); 2Department of Orthopedic Surgery, Okazaki City Hospital, Okazaki 444-8553, Japan; amlastregret@gmail.com

**Keywords:** T1 slope, C2-7 SVA, C2-7 angle

## Abstract

Introduction: The T1 slope is important for cervical surgical planning, and it may be invisible on radiographic images. The prevalence of T1 invisible cases and the differences in demographic and radiographic characteristics between patients whose T1 slopes are visible or invisible remains unexplored. Methods: This pilot study aimed to evaluate the differences in these characteristics between outpatients whose T1 slopes were visible or invisible on radiographic images. Patients (*n* = 60) who underwent cervical radiography, whose T1 slope was confirmed clearly, were divided into the visible (V) group and invisible (I) group. The following radiographic parameters were measured: (1) C2-7 sagittal vertical axis (SVA), (2) C2-7 angle in neutral, flexion, and extension positions. Results: Based on the T1 slope visibility, 46.7% of patients were included in group I. The I group had significantly larger C2-7 SVA than the V group for males (*p* < 0.05). The C2-7 SVA tended to be larger in the I group, without significant difference for females (*p* = 0.362). Discussion: The mean C2-7 angle in neutral and flexion positions was not significantly different between the V and I groups for either sex. The mean C2-7 angle in the extension position was greater in the V group. The T1 slope was invisible in males with high C2-7 SVA.

## 1. Introduction

The T1 slope is defined as the angle between the horizontal line and superior endplate of the T1 vertebra [1]. It has been used to evaluate the sagittal balance of the cervical spine and has been reported to have a strong correlation with greater sagittal malalignment of the dens [1]. T1 slope angle, neck tilt, and thoracic inlet angle have been reported as significant cervical sagittal parameters, similar to the concept that pelvic incidence, pelvic tilt, and lumbar lordosis are important lumbosacral parameters in patients with adult spinal deformity [2,3,4,5,6]. The relationship between health-related quality of life and surgical outcomes and T1 slope has been examined in several studies [7,8].

T1 slope minus cervical lordosis can predict ideal cervical lordosis, and T1 slope plays an important role in planning cervical surgery: predicting the progression of kyphosis after cervical laminoplasty and the ideal correction angle in posterior cervical instrumentation [2,3,9]. However, the T1 vertical body is not always clearly visible on lateral cervical radiographic images because of interference by the shoulder and thoracic trunk in obese and short-necked patients [10,11]. Qiao et al. reported that the T1 upper endplate had poor visibility in 34% of cases with plain X-ray radiographs [12]. In such cases, appropriate surgical planning and clinical studies excluding cases with invisible T1 slopes have a selection bias and can be challenging.

However, to the best of our knowledge, the percentage of patients with invisible T1 slopes and the difference in radiological characteristics between patients with visible or invisible T1 slopes remain unknown [13]. Therefore, this pilot study aimed to investigate the differences between these characteristics in outpatients with visible or invisible T1 slopes.

## 2. Materials and Methods

### 2.1. Study Design and Ethics Approval

Our study retrospectively included adult patients with neck pain who underwent lateral radiography with their necks in neutral, flexion, and extension positions from 2015 to 2016. The reason for radiography included spinal degenerative diseases, spinal tumors, and ossification of the posterior longitudinal ligament. Patients with a history of cervical infection, fractures, or surgery were excluded. This study was approved by the Research Ethics and Conflicts Committee of our University and was performed in accordance with the Declaration of Helsinki.

### 2.2. Radiological Assessment

Cervical lateral radiographs were obtained using standard radiographic techniques, in which the tube was centered on the level of the center of the xiphoid process. Lateral radiographic images were obtained with each participant standing and looking straight ahead. Flexion and extension radiographs were obtained with the neck in maximum flexion and extension. T1 slope angles were measured using these images, with the T1 slope defined as the superior endplate of the T1 vertebrae [12]. Three spine surgeons evaluated the visibility of the T1 slope. In cases where the surgeons disagreed regarding the visibility of the T1 slope, a discussion was held to reach a conclusion. Patients were divided into two groups based on whether the T1 slope was visible (V) or invisible (I), as decided by the three surgeons.

In this study, we used dynamic range control processing methods to improve the clarity of the radiographs. Dynamic range control processing can change the density and contrast of only low- and high-density areas [14].

The measured parameters in the radiographs were as follows: the Cobb angle from C2 to C7 (C2-7 angle) was defined as the angle between the inferior endplates of C2 and C7 on standing lateral radiographs. The C2-7 angle was measured in neutral, flexion, and extension positions. The C2-7 sagittal vertical axis (SVA) was defined as the deviation of the C2 plumb line (extending from the centroid of the C2 vertebra) from the superior posterior endplate of C7, with positive sagittal alignment defined as an anterior deviation. All parameters were measured twice by the same researcher independently using the same method.

### 2.3. Statistical Analyses

Sex was compared between the V and I groups using the Chi-square test. Patient age, body mass index (BMI), C2-7 SVA, and C2-7 angle measurements on radiographs were compared between the V and I groups using the Student’s t-test. Each analysis was performed separately for men and women. For data aggregation and analyses, we used the IBM SPSS Statistics version 24.0 software (IBM Corp., Armonk, NY, USA), and *p* < 0.05 was considered statistically significant.

## 3. Results

The study population consisted of 60 consecutive patients with cervical spine disorders other than cervical spine trauma who visited our hospital between 2015 and 2016; of these, 30 (50.0%) were female, and the average age was 44.5 years (range: 34–81 years).

Among the 60 patients, 53.3% (32 patients; 10 men and 22 women) were included in the V group, and 46.7% (28 patients; 20 men and eight women) were included in the I group (Table 1, Figure 1).

Among the 60 patients, there were spinal degenerative diseases (*n* = 28), spinal tumors (*n* = 9), ossification of the posterior longitudinal ligament (*n* = 14), and other conditions (*n* = 9). The V group included 16 patients with spinal degenerative diseases, 6 with spinal tumors, 10 with ossification of the posterior longitudinal ligament, and 5 with other conditions. The I group included 10 patients with spinal degenerative diseases, 4 with spinal tumors, 6 with ossification of the posterior longitudinal ligament, and 5 with other conditions. There was no significant difference between the V and I groups (*p* = 0.943) (Table 2).

Of the 60 patients, 23 and 37 received surgical and conservative treatment, respectively. In the V group, 12 patients received surgical treatment while 20 received conservative treatment. The I group included 11 patients who received surgical treatment and 17 who received conservative treatment. There was no significant difference between the V and I groups (*p* = 0.887) (Table 3).

There were significantly more males in the I group (*p* < 0.05) (Table 3). No significant differences were observed between the groups regarding age or BMI (Table 4). The mean age was 44.4 years (range: 35–78 years) in the V group, and 44.6 years (range: 34–81) in the I group. The mean BMI was 21.2 (range: 19.6–24.3) and 22.1 (range: 19.9–26.1) in the V and I groups, respectively.

We compared the C2-7 SVA and C2-7 angles between the V and I groups for each sex. The mean C2-7 SVA was 16.0 mm in the V group and 28.9 mm in the I group for males. The I group had significantly greater C2-7 SVA than the V group (*p* < 0.05) (Figure 2). For females, the mean C2-7 SVA in the V group was 19.9 mm and 24.4 mm in the I group. Similar to males, there was a higher C2-7 SVA in the I group, but there was no significant difference (*p* = 0.362) (Figure 2).

In the neutral position, the mean C2-7 angle in males was 11.8° in the V group, and 12.1° in the I group, and the mean C2-7 angle in females was 11.0° and 10.4° in the V and I groups, respectively. There was no difference in the C2-7 angle in the neutral position for both males and females between the V and I groups (Figure 3).

Similar to the neutral position, there was no significant difference in the flexion position. In the flexion position, the mean C2-7 angle in males was −14.8° in the V group, and −14.0° in the I group, and the mean C2-7 angle in females was −9.4° in the V group and −8.7° in the I group. There was no significant difference in the flexion position in females: 39.2° in the V group and 30.7° in the I group (*p* = 0.147) (Figure 3). In contrast, in the extension position, the mean C2-7 angle in males was greater in the V group: 37.6° and 24.4° in the V and I groups, respectively (*p* < 0.05) (Figure 3).

Thus, male patients with greater C2-7 SVA had an invisible T1 slope, as shown in the representative cases in Figure 4A,B.

In this study, some surgical cases were included. There was no clear difference in the imaging changes in surgical cases between both groups before and one year after surgery (Figure 5).

## 4. Discussion

To the best of our knowledge, this is the first study comparing the characteristics of patients with and without an identifiable T1 slope. This was a pilot study in outpatients as a preliminary step to determine the clinical significance of T1 slope visibility on surgery. In this study, the T1 slope could not be identified in 46.7% of the cases. T1 slope was invisible predominantly in males. In males, the C2-7 SVA was significantly larger, and the C2-7 angle in extension was significantly smaller in cases with an invisible T1 slope. The same trend was observed in females, but the difference was insignificant. Thus, the T1 slope was invisible in the male physique, and cases with anterior cervical shift characterized by larger C2-7 SVA.

The T1 slope was first reported by Knott et al. [1] in 2010 as the T1 sagittal angle. They noted that the T1 slope was positively correlated with the C2 SVA, influencing the global sagittal balance. Additionally, several other recent studies have addressed the relationship between the T1 slope and other parameters of the global sagittal balance of the spine. Patients with a large T1 slope require large cervical lordosis to preserve the sagittal balance of the cervical spine, suggesting that the T1 slope affects cervical sagittal alignment [14]. Furthermore, Lee et al. [3,15] reported a significant relationship between the T1 slope and thoracic kyphosis. A significant weak correlation between the T1 slope and lumbar lordosis was also reported [15]. These findings suggest that the T1 slope is associated with cervical sagittal alignment and thoracic and lumbar sagittal alignment, in addition to the fact that the T1 slope is an important factor for global sagittal alignment. However, it has been reported that upright cervical radiographs are more accurate than whole spine radiographs for evaluating cervical spine alignment, including T1 slope. This study used upright cervical spine radiographs instead of whole spine radiographs [16].

Despite the importance of the T1 slope, the T1 vertebral body is often unclear on cervical radiographic images due to interference of the shoulder contour, especially in obese or muscular individuals [11,17], resulting in difficulty in identifying the T1 slope. In previous reports, the T1 slope was difficult to identify in approximately one-third of the cases [12,17]. In this study, the T1 slope could not be identified in 46.7% of the cases, almost consistent with the previous observation.

Several alternative methods have been investigated in cases with an invisible T1 slope, including magnetic resonance imaging (MRI) or computed tomography (CT). T1s-CL is important for cervical spine postoperative alignment assessment. In a study examining alternative parameters to T1s-CL, the C2 slope was the parameter that correlated best with pre- and postoperative changes [18]. Supine MRI and CT images cannot be considered an alternative to the T1 slope on lateral radiographs as these images are not captured in the upright position [19]. Jun et al. reported that the T1 slope angle on radiographs was larger than on CT images. However, a significant correlation was noted between the T1 slope angles on radiographs and CT images [10]. Park et al. correlated C7 slope with T1 slope by measuring T1 slope on CT based on C7 slope on X-ray. However, they excluded cases where the T1 slope was not visible on the X-ray [13]. A strong correlation between C7 slope and T1 slope has been found in MRI studies with the patient seated in upright weight-bearing neutral positions. However, no comparison with x-rays has been made [20]. Ideally, the imaging modality enabling clear visualization of the T1 upper endplate in an upright position, such as EOS^®^ (EOS^®^ imaging, Paris, France), is generally desired [21]. EOS^®^ is the EOS imaging system, a novel technique that allows for acquiring images of the body or of body sections in standing position and under normal weight-bearing conditions [22]. However, many facilities do not have access to EOS, and researchers have to exclude cases in which the T1 upper endplate is invisible. As an alternative to the T1 slope on lateral radiographs, the C7 slope reportedly correlated significantly with the T1 slope [12,17].

We showed that cases with invisible T1 slopes had some notable characteristics. First, the T1 slope was invisible in males. This may be affected by differences in body thickness and shoulder position between men and women: the T1 upper endplate could be affected by the shoulder and thoracic trunk, especially in obese and short-necked patients [10]. Furthermore, Reynolds et al. demonstrated that neck circumference influences cervical sagittal alignment [23]. Neck circumference is influenced by muscle or obesity [24]; thus, it can influence T1 slope visibility. Further, we found significant differences in extension (C2-7) angle and C2-7 SVA between patients in the I and V groups. In cases where the cervical vertebrae shift forward and the cervical backward bending angle is small, T1 is hidden under the soft tissue and appears invisible, as shown in Figure 4. The risk of kyphotic deformity after cervical decompression is high in cases with a large C2-7 SVA shift [25]. Many cases require deformity correction, and the T1 slope must be considered to achieve an ideal cervical lordosis. The classification of cervical spine deformity is based on the T1 slope [26]. However, it might be necessary to consider a new classification based on a reliably measurable index such as the C7 slope [12].

There are some limitations to this study. First, the number of patients included in this study was small. Further, this study was conducted on symptomatic patients, which may have affected the global alignments. To overcome this problem, clinicians should consider investigating the radiography of healthy individuals. However, these patients may be evaluated in clinical studies for surgical results. Furthermore, the condition of radiographic imaging was not completely unified; thus, it could affect T1 visibility. It would also have been desirable for the three surgeons to blindly classify the patients into the V and I groups before starting the study. However, since there were no numerical data, the three surgeons discussed the decision from the beginning. As a result, it was necessary to discuss and decide one way or the other, and the result would not have been different even if we had done it blindly at the beginning. In addition, C2-7 SVA and C2-7 angles were measured by a single person, and the validation is insufficient.

## 5. Conclusions

In summary, our study analyzed the differences in demographic and radiological characteristics between patients with visible or invisible T1 slopes. Our findings suggest that the T1 slope tends to be invisible in males with greater C2-7 SVA. This was a pilot study. Therefore, based on the results of this study, we would like to accumulate surgical cases in the future and further investigate the clinical significance of this study.

## Figures and Tables

**Figure 1 jcm-11-00411-f001:**
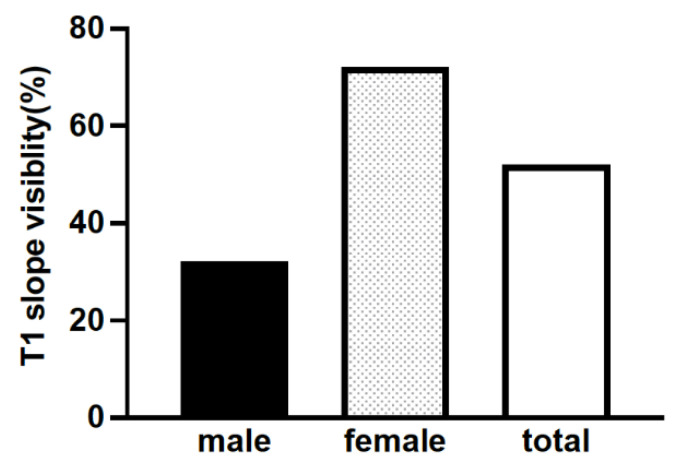
T1 slope visibility.

**Figure 2 jcm-11-00411-f002:**
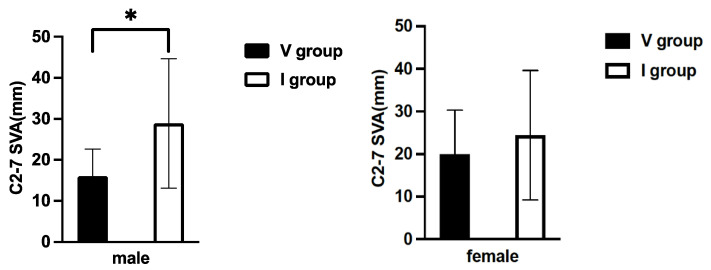
Comparison of the C2-7 SVA between the I and V groups in males and females. *: *p* < 0.05; V group: T1 slope visible; I group: T1 slope invisible; C2-7 SVA, C2-7 sagittal vertical axis.

**Figure 3 jcm-11-00411-f003:**
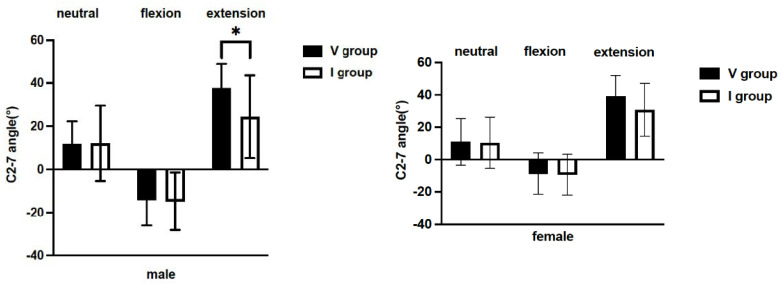
Comparison of the C2-7 angles between the I and V groups in neutral, flexion, and extension in males and females. *: *p* < 0.05; V group, T1 slope visible; I group, T1 slope invisible.

**Figure 4 jcm-11-00411-f004:**
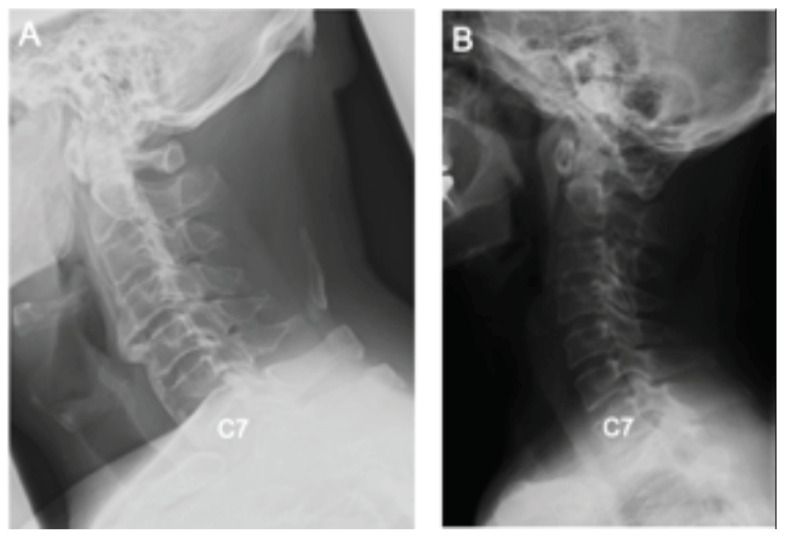
Comparison between invisible and visible T1 slopes of patients. (**A**) Patient with invisible T1 slope: 62 years old, male, ossification of the posterior longitudinal ligament, C2-7 SVA 58 mm, C2-7 angle 4°. (**B**) Patient with visible T1 slope: 63 years old, female, ossification of the posterior longitudinal ligament, C2-7 SVA 19 mm, C2-7 angle 22°.

**Figure 5 jcm-11-00411-f005:**
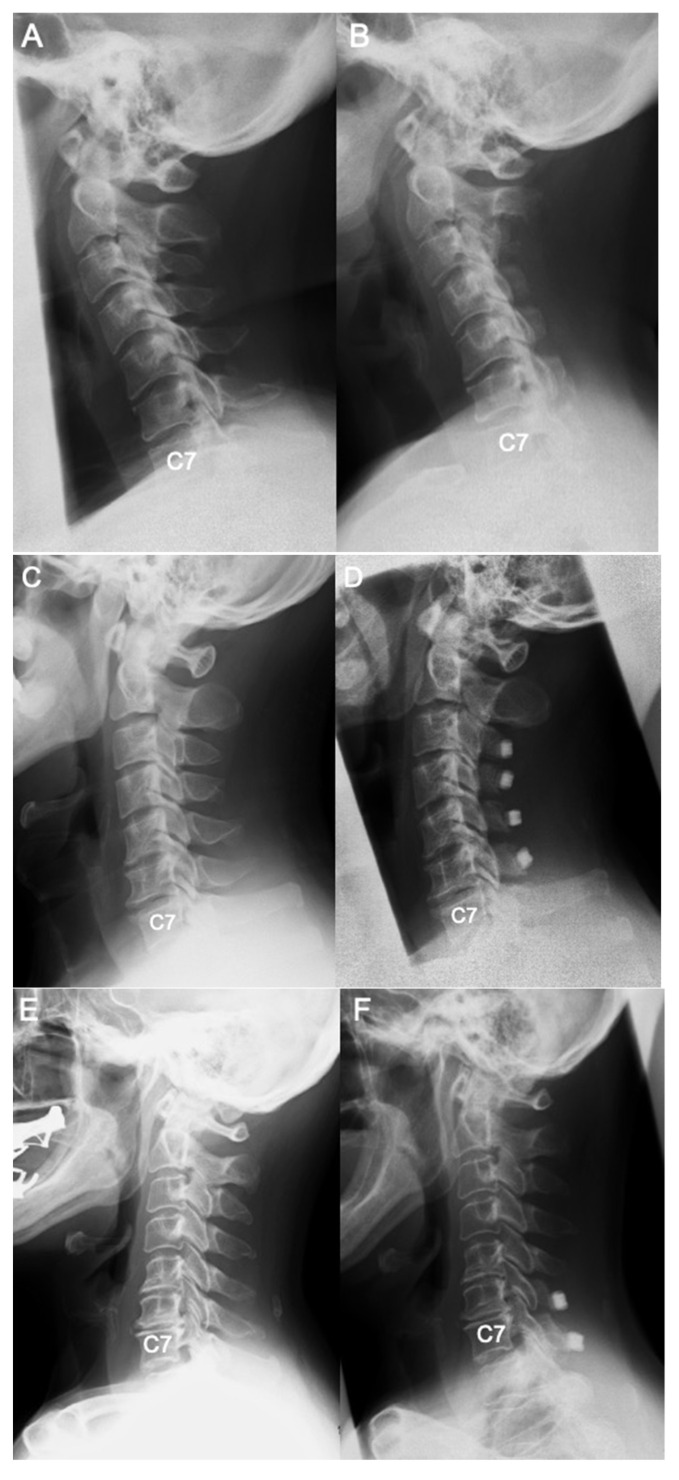
Pre and postoperative radiographs of patients with visible and invisible T1 slope. (**A**,**B**) Patient with invisible T1 slope: 50 years old, female, ossification of the posterior longitudinal ligament, C2-7 laminoplasty A; the preoperative radiograph, C2-7 SVA 39 mm, C2-7 angle 1°, C2 slope 27°, C7 slope 22°, B; the radiograph 1 year after surgery, C2-7 SVA 37 mm, C2-7 angle 11°, C2 slope 27°, C7 slope 17°. (**C**,**D**) Patient with invisible T1 slope: 51 years old, male, cervical spondylotic myelopathy, C3-7 laminoplasty A; the preoperative radiograph, C2-7 SVA 21 mm, C2-7 angle 15°, C2 slope 11°, C7 slope 22°, B; the radiograph 1 year after surgery, C2-7 SVA 18 mm, C2-7 angle 25°, C2 slope 5°, C7 slope 24. (**E**,**F**) Patient with visible T1 slope: 62 years old, female, ossification of the posterior longitudinal ligament, C5-7 laminoplasty A; the preoperative radiograph, C2-7 SVA 4 mm, C2-7 angle 7°, C2 slope 6°, C7 slope 7°, T1 slope 6°, B; the radiograph 1 year after surgery, C2-7 SVA 14 mm, C2-7 angle 2°, C2 slope 14°, C7 slope 16°, T1 slope 14°. (**G**,**H**) Patient with visible T1 slope: 44 years old, male, cervical spondylotic myelopathy, C3-7 laminoplasty A; the preoperative radiograph, C2-7 SVA 8 mm, C2-7 angle 12°, C2 slope 11°, C7 slope 17°, T1 slope 24°, B; the radiograph 1 year after surgery, C2-7 SVA 21 mm, C2-7 angle 1°, C2 slope 23°, C7 slope 18°, T1 slope 20°.

**Table 1 jcm-11-00411-t001:** T1 slope visibility.

	Total (*n* = 60)	Male (*n* = 30)	Female (*n* = 30)	*p*-Value
T1 slope visibility (V group/I group)	53.3% (32/28)	33.3% (10/20)	73.3% (22/8)	<0.05

**Table 2 jcm-11-00411-t002:** Cervical disorders in patients (*n* = 60).

	No. of Patients		
Disorder	Total	V Group	I Group
Degenerative disorders	28	15	13
Spinal cord tumors	9	5	4
OPLL	14	8	6
Others	9	4	5

OPLL = ossification of the posterior longitudinal ligament. There was no significant difference between the V and I groups (*p* = 0.943).

**Table 3 jcm-11-00411-t003:** Treatment for patients (*n* = 60).

	No. of Patients		
Treatment	Total	V Group	I Group
Surgical treatment	23	12	11
Conservative treatment	37	20	17

Surgical treatment: decompression, posterior fusion, anterior fusion, resection of tumors. Conservative treatment: medication, rehabilitation, observation. There was no significant difference between the V and I groups (*p* = 0.887).

**Table 4 jcm-11-00411-t004:** Baseline demographic characteristics of patients.

	Total	Visible (V group)	Invisible (I Group)	*p*-Value
Age (years)	44.5 ± 11.1	44.4 ± 12.5	44.6 ± 8.3	n.s.
BMI	21.6 ± 2.4	21.2 ± 2.3	22.1 ± 2.5	n.s.
Male:Female	30:30	10:22	20:8	<0.05

Values presented as mean ± SD; n.s., not significant; BMI, body mass index.

## Data Availability

The data of this study are available from the corresponding author upon request.

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
