# Peer review of "Differences in Demographic and Radiographic Characteristics between Patients with Visible and Invisible T1 Slopes on Lateral Cervical Radiographic Images"

_jcm, 2022, doi:10.3390/jcm11020411_

Round 1

Reviewer 1 Report

Evaluation

  1. The abstract is sufficiently informative and the content is consistent with the rest of the paper.

  2. The introduction provides sufficient background information on the research topic, the research

    question and hypothesis are identifiable. But this is an insufficient study design for the study

    aims.

  3. Appropriate methods were used to answer the research question. The authors mentioned to

  4. The results are very short and they are not described with sufficient detail

  5. I had difficulties in interpreting the tables.

  6. The discussion is clear, but it should also address the limitations of the study.

    The conclusions are an insufficient interpretation of the results.
    The discussion shows the relationship of the study to previous research, but does not include practical applications.

  7. The references are not formatted correctly.

Author Response

Evaluation

The abstract is sufficiently informative and the content is consistent with the rest of the paper.

The introduction provides sufficient background information on the research topic, the research question and hypothesis are identifiable. But this is an insufficient study design for the study aims.

Appropriate methods were used to answer the research question.

The authors mentioned to

The results are very short and they are not described with sufficient detail. I had difficulties in interpreting the tables.

Thank you for your comments.

I changed the results and added some details as attached file.

The discussion is clear, but it should also address the limitations of the study.

I added the following to the limitation .

There are some limitations to this study. First, the number of patients included in this study was small. Second, this study was conducted on symptomatic patients, which may have affected the global alignments. To overcome this problem, clinicians should consider investigating the radiography of healthy individuals. However, these patients may be evaluated in clinical studies for surgical results. Third, the condition of radiographic imaging was not completely unified; thus, it could affect T1 visibility. Forth, it would have been desirable for the three surgeons to blindly classify the patients into V group and I group before starting the study, but since there was no numerical data, the three surgeons discussed the decision from the beginning. As a result, it was necessary to discuss and decide one way or the other, and the result would not have been different even if we had done it blindly at the beginning. In addition, C2-7 SVA and C2-7 angle are measured by a single measurer, and the validation is insufficient.

The conclusions are an insufficient interpretation of the results.
The discussion shows the relationship of the study to previous research, but does not include practical applications.

Thank you for pointing this out.

It is a very important part and we are most concerned about the effect on the surgical outcome.

However, this study is not limited to surgical cases, but all patients who visited the outpatient clinic.

Surgical cases are only a part of the total number of cases, and we do not have enough cases to make comparisons.

We have added a few cases to the results. We have not been able to find any definite factors affecting surgery in this study.

We hope to clarify this in further studies in the future.

This is a pilot study. So we changed conclusions as follows;

In summary, our study analyzed the differences in demographic and radiological characteristics between patients with visible or invisible T1 slopes. Our findings suggest that the T1 slope tends to be invisible in males with greater C2-7 SVA. This is a pilot study. Therefore, based on the results of this study, we would like to accumulate surgical cases in the future and further investigate the clinical significance of this study.

The references are not formatted correctly.

We corrected references.

Reviewer 2 Report

Authors presented a retrospective review on 60 patients who underwent cervical radiography,  divided into the visible (V) group and invisible (I) group(46.7%) regarding the T1 slope. Patients with invisible T1 slope had larger C2-7 SVA than the V group for males, the mean C2-7 angle in neutral and flexion positions was not significantlydifferent between the groups and the mean C2-7 angle in the extension position was greater in the visible group.

I suggest to expand the Introductiono on clinical relevance of T1 slope measurment. Several important clinical data on the patients who were included in the study are missing. I suggest to make a table and incorporate all patients with gender, age, diagnosis and the surgical treatment.

It would be interesting to explain - based on several illustrative cases for each disease (degenerative, spinal cord tumors, OPLL, others) how did the T1 slope (in)visibility affect the surgical treatment. This could make this manuscript more clinically relevant, as at the moment this is merely a radiographic study. Please provide for at least three cases pre- and postoperative radiographs with measurments of the T1 slope.

If possible, perform measurments of the T1 slope and C2-7 in the same patients following treatment and compare for differences between visible and invisible T1 slope group.

T1 slope-cervical lordosis with C2 slope seems to be more important for planning cervical surgery than T1 slope alone. Furthermore, C7 slope shows better visibility than T1 slope. Include these parameters in the final assesment for better validation of the results.

For Discussion please include following studies and comment:

  1. Lee HJ, You ST, Sung JH, Kim IS, Hong JT. Analyzing the Significance of T1 Slope minus Cervical Lordosis in Patients with Anterior Cervical Discectomy and Fusion Surgery. J Korean Neurosurg Soc. 2021;64(6):913-921. doi:10.3340/jkns.2021.0011
  2. Park BJ, Gold CJ, Woodroffe RW, Yamaguchi S. What is the most accurate substitute for an invisible T1 slope in cervical radiographs? A comparative study of a novel method with previously reported substitutes. J Neurosurg Spine. 2021 Nov 26:1-7. doi: 10.3171/2021.8.SPINE21901. Epub ahead of print. PMID: 34826812.
  3. Lee DH, Park S, Kim DG, Hwang CJ, Lee CS, Hwang ES, Cho JH. Cervical spine lateral radiograph versus whole spine lateral radiograph: A retrospective comparative study to identify a better modality to assess cervical sagittal alignment. Medicine (Baltimore). 2021 May 28;100(21):e25987. doi: 10.1097/MD.0000000000025987. PMID: 34032714; PMCID: PMC8154400.
  4. Tamai K, Buser Z, Paholpak P, Sessumpun K, Nakamura H, Wang JC. Can C7 Slope Substitute the T1 slope?: An Analysis Using Cervical Radiographs and Kinematic MRIs. Spine (Phila Pa 1976). 2018 Apr 1;43(7):520-525. doi: 10.1097/BRS.0000000000002371. PMID: 28767624

Author Response

Authors presented a retrospective review on 60 patients who underwent cervical radiography, divided into the visible (V) group and invisible (I) group(46.7%) regarding the T1 slope. Patients with invisible T1 slope had larger C2-7 SVA than the V group for males, the mean C2-7 angle in neutral and flexion positions was not significantly different between the groups and the mean C2-7 angle in the extension position was greater in the visible group. 

I suggest to expand the Introductiono on clinical relevance of T1 slope measurment. Several important clinical data on the patients who were included in the study are missing. I suggest to make a table and incorporate all patients with gender, age, diagnosis and the surgical treatment.

Thank you for your comments.

I changed introduction as follows.

I added all patients’ data to Table 2 and 3.

The T1 slope has been used to evaluate the sagittal balance of the cervical spine and has been reported to have a strong correlation with greater sagittal malalignment of the dens [1]. The T1 slope is defined as the angle between the horizontal line and superior endplate of the T1 vertebra [1]. T1 slope angle, neck tilt, and thoracic inlet angle have been reported as significant cervical sagittal parameters, similar to the concept that pelvic incidence, pelvic tilt, and lumbar lordosis are important lumbosacral parameters in patients with adult spinal deformity [2,3]. The relationship between health-related quality of life (HRQOL) and surgical outcomes and T1 slope has been examined in a number of studies [4,5].

Table 2 Cervical disorders in patients (n=60)

No. of Patients

Disorder

total

V group

I group

Degenerative disorders

28

15

13

Spinal cord tumors

9

5

4

OPLL

14

8

6

Others

9

4

5

OPLL = ossification of the posterior longitudinal ligament

There was no significant difference between the V and I groups (p=0.943).

Table3 Baseline demographic characteristics of patients

total

Visible (V group)

Invisible (I group)

p-value

Age (years)

44.5±11.1

44.4±12.5

44.6±8.3

n.s.

BMI

21.6±2.4

21.2±2.3

22.1±2.5

n.s.

Male : Female

30:30

10:22

20:8

<0.05

Values presented as mean ± SD, n.s.: not significant; BMI: body mass index

It would be interesting to explain - based on several illustrative cases for each disease (degenerative, spinal cord tumors, OPLL, others) how did the T1 slope (in)visibility affect the surgical treatment. This could make this manuscript more clinically relevant, as at the moment this is merely a radiographic study. Please provide for at least three cases pre- and postoperative radiographs with measurments of the T1 slope.

Thank you for your comments.

The results are not comparable because they are based on outpatients and not all cases are surgical cases, but we think they are very important findings.

We have added the results of some of the surgical cases as attached files. We have not found a clear difference between the V group and the I group in terms of pre- and post-operative changes based on the results of this study alone.

If possible, perform measurments of the T1 slope and C2-7 in the same patients following treatment and compare for differences between visible and invisible T1 slope group.

Thank you very much for your valuable comments.

The data presented here are for outpatients, and only a small number of surgical cases were included, so we were unable to examine the surgical outcomes due to insufficient data.

T1 slope-cervical lordosis with C2 slope seems to be more important for planning cervical surgery than T1 slope alone. Furthermore, C7 slope shows better visibility than T1 slope. Include these parameters in the final assesment for better validation of the results.

I think you are absolutely right. However, a comparative paper on C7 has been published by our group, and we have already shown its validity, so we did not consider it in this study.

Inoue, T.; Ando, K.; Kobayashi, K.; Nakashima, H.; Ito, K.; Katayama, Y.; Machino, M.; Kanbara, S.; Ito, S.; Yamaguchi, H.; et al. Age-related Changes in T1 and C7 Slope and the Correlation Between Them in More Than 300 Asymptomatic Subjects. Spine 2021, 46.

For Discussion please include following studies and comment:

  1. Lee HJ, You ST, Sung JH, Kim IS, Hong JT. Analyzing the Significance of T1 Slope minus Cervical Lordosis in Patients with Anterior Cervical Discectomy and Fusion Surgery. J Korean Neurosurg Soc. 2021;64(6):913-921. doi:10.3340/jkns.2021.0011
  2. Park BJ, Gold CJ, Woodroffe RW, Yamaguchi S. What is the most accurate substitute for an invisible T1 slope in cervical radiographs? A comparative study of a novel method with previously reported substitutes. J Neurosurg Spine. 2021 Nov 26:1-7. doi: 10.3171/2021.8.SPINE21901. Epub ahead of print. PMID: 34826812.
  3. Lee DH, Park S, Kim DG, Hwang CJ, Lee CS, Hwang ES, Cho JH. Cervical spine lateral radiograph versus whole spine lateral radiograph: A retrospective comparative study to identify a better modality to assess cervical sagittal alignment. Medicine (Baltimore). 2021 May 28;100(21):e25987. doi: 10.1097/MD.0000000000025987. PMID: 34032714; PMCID: PMC8154400.
  4. Tamai K, Buser Z, Paholpak P, Sessumpun K, Nakamura H, Wang JC. Can C7 Slope Substitute the T1 slope?: An Analysis Using Cervical Radiographs and Kinematic MRIs. Spine (Phila Pa 1976). 2018 Apr 1;43(7):520-525. doi: 10.1097/BRS.0000000000002371. PMID: 28767624

Thank you for your comments.

I changed disucussion as follows;

The T1 slope was first reported by Knott et al. [1] in 2010 as the T1 sagittal angle. They noted that the T1 slope was positively correlated with the C2 SVA, which influenced the global sagittal balance. Additionally, several recent studies have addressed the relationship between the T1 slope and other parameters of the global sagittal balance of the spine. Patients with a large T1 slope require large cervical lordosis to preserve the sagittal balance of the cervical spine, suggesting that the T1 slope affects cervical sagittal alignment [10]. Furthermore, Lee et al. [3,11] reported a significant relationship between the T1 slope and thoracic kyphosis. In addition, a significant weak correlation between the T1 slope and lumbar lordosis was also reported [11]. These findings suggest that the T1 slope is associated with cervical sagittal alignment as well as thoracic and lumbar sagittal alignment, in addition to the fact that the T1 slope is an important factor for global sagittal alignment. However, it has been reported that upright cervical radiographs are more accurate than whole spine for evaluating cervical spine alignment, including T1slope. In this study, we used upright cervical spine radiographs instead of whole spine radiographs[12].

Despite the importance of the T1 slope, the T1 vertebral body is often unclear on cervical radiographic images due to interference of the shoulder contour, especially in obese or muscular individuals [8,13], resulting in difficulty in identifying the T1 slope. In previous reports, the T1 slope was difficult to identify in approximately one-third of the cases [9,13]. In this study, the T1 slope could not be identified in 46.7% of the cases, which was almost consistent with the previous observation.

In cases with an invisible T1 slope, several alternative methods have been investigated, including magnetic resonance imaging (MRI) or computed tomography (CT). T1s-CL is important for cervical spine postoperative alignment assessment; in the study examining alternative parameters to T1S-CL, C2S was the parameter that correlated best with pre- and postoperative changes [14]. Supine MR and CT images cannot be considered an alternative to the T1 slope on lateral radiographs as these images are not captured in the upright position [15]. Jun et al. reported that the T1 slope angle on radiographs was larger than that on CT images, although a significant correlation was noted between the T1 slope angles on radiographs and CT images [7]. Park BJ et al correlated C7slope with T1slope by measuring T1slope on CT based on C7slope on x-ray. However, they excluded cases in which the T1slope was not visible on x-ray [16]. A strong correlation between C7slope and T1slope has been found within MRI in studies of MRI with the patient seated in upright weight-bearing neutral positions.. However, no comparison with x-rays has been made [17]. Ideally, the imaging modality enabling clear visualization of the T1 upper endplate in an upright position, such as EOS (the EOS imaging system by EOS imaging which is a novel technique which allows for the acquirement of images of the body or of body sections in standing position and under normal weight-bearing conditions) is generally desired [18]; however, many facilities do not have access to EOS, and researchers have to exclude cases in which the T1 upper endplate is invisible. As an alternative to the T1 slope on lateral radiographs, the C7 slope reportedly correlated significantly with the T1 slope [9,13].

Reviewer 3 Report

English grammar and composition was outstanding and without error.

The introduction is comprehensive but concise and transitions well into the methods section. This reviewer is concerned about a study whose aims are to contribute generalized knowledge, the investigators have used a very small sample size. If the authors' goal was only to produce pilot data, then this reviewer would find the sample size adequate. However, if the intended goal is to perform a pilot study, the authors should clearly state this intention.. Moreover, a retrospective analysis is more conducive to preliminary pilot data analysis as well. There are also some concerns about the study involving a relatively homogenous population at a single institution and the effects this could have on study outcomes, which would be of less concern if this is only a pilot study. This reviewer has some concerns about the methods used for radiograph evaluations and angle measurement. 

First, it would have been better to have the 3 spine surgeons independently decide if the T1 slope was visible and to be blinded to the opinions of the other 2 surgeons. In this case, a 4th person could have tabulated the outcomes of these surgeons' determinations. This reviewer suggests that disagreement among the surgeons might have been better mediated with the assistance of radiologists experienced in this subspecialty.

Second, the reviewer is concerned that only one investigator was responsible for all angle measurements. Again, this would be independently determined by multiple blinded reviewers and aggregated by another individual, and mediated by experienced radiologists in this subspeciality.

Regarding statistical analysis, this reviewer questions why the investigators only performed tests of difference and not tests of correlation.  EOS is not defined.

Author Response

The introduction is comprehensive but concise and transitions well into the methods section. This reviewer is concerned about a study whose aims are to contribute generalized knowledge, the investigators have used a very small sample size. If the authors' goal was only to produce pilot data, then this reviewer would find the sample size adequate. However, if the intended goal is to perform a pilot study, the authors should clearly state this intention.. Moreover, a retrospective analysis is more conducive to preliminary pilot data analysis as well. There are also some concerns about the study involving a relatively homogenous population at a single institution and the effects this could have on study outcomes, which would be of less concern if this is only a pilot study. This reviewer has some concerns about the methods used for radiograph evaluations and angle measurement. 

Thank you for your comments.

We added to conclusion as follows;

In summary, our study analyzed the differences in demographic and radiological characteristics between patients with visible or invisible T1 slopes. Our findings suggest that the T1 slope tends to be invisible in males with greater C2-7 SVA. This is a pilot study. Therefore, based on the results of this study, we would like to accumulate surgical cases in the future and further investigate the clinical significance of this study.

First, it would have been better to have the 3 spine surgeons independently decide if the T1 slope was visible and to be blinded to the opinions of the other 2 surgeons. In this case, a 4th person could have tabulated the outcomes of these surgeons' determinations. This reviewer suggests that disagreement among the surgeons might have been better mediated with the assistance of radiologists experienced in this subspecialty.

We think you are right. It would have been ideal to do a blind evaluation first and then discuss the final results. We have added this information to the limitations. It is also currently difficult to get a fourth surgeon or radiologist to do the evaluation.

There are some limitations to this study. First, the number of patients included in this study was small. Second, this study was conducted on symptomatic patients, which may have affected the global alignments. To overcome this problem, clinicians should consider investigating the radiography of healthy individuals. However, these patients may be evaluated in clinical studies for surgical results. Third, the condition of radiographic imaging was not completely unified; thus, it could affect T1 visibility. Forth, it would have been desirable for the three surgeons to blindly classify the patients into V group and I group before starting the study, but since there was no numerical data, the three surgeons discussed the decision from the beginning. As a result, it was necessary to discuss and decide one way or the other, and the result would not have been different even if we had done it blindly at the beginning. In addition, C2-7 SVA and C2-7 angle are measured by a single measurer, and the validation is insufficient.

Second, the reviewer is concerned that only one investigator was responsible for all angle measurements. Again, this would be independently determined by multiple blinded reviewers and aggregated by another individual, and mediated by experienced radiologists in this subspeciality.

Thanks for the great advice. You are right. We will add this to the limitation as it is difficult to cooperate with additional measurements.

There are some limitations to this study. First, the number of patients included in this study was small. Second, this study was conducted on symptomatic patients, which may have affected the global alignments. To overcome this problem, clinicians should consider investigating the radiography of healthy individuals. However, these patients may be evaluated in clinical studies for surgical results. Third, the condition of radiographic imaging was not completely unified; thus, it could affect T1 visibility. Forth, it would have been desirable for the three surgeons to blindly classify the patients into V group and I group before starting the study, but since there was no numerical data, the three surgeons discussed the decision from the beginning. As a result, it was necessary to discuss and decide one way or the other, and the result would not have been different even if we had done it blindly at the beginning. In addition, C2-7 SVA and C2-7 angle are measured by a single measurer, and the validation is insufficient.

Regarding statistical analysis, this reviewer questions why the investigators only performed tests of difference and not tests of correlation.  EOS is not defined.

Thank you for your comments.

In this study, we focused on whether T1S was visible or not, so we compared the presence or absence of T1S rather than the correlation.

The EOS was not fully explained. We have added the following information.

Ideally, the imaging modality enabling clear visualization of the T1 upper endplate in an upright position, such as EOS (the EOS imaging system by EOS imaging which is a novel technique which allows for the acquirement of images of the body or of body sections in standing position and under normal weight-bearing conditions) is generally desired [18]; however, many facilities do not have access to EOS, and researchers have to exclude cases in which the T1 upper endplate is invisible. As an alternative to the T1 slope on lateral radiographs, the C7 slope reportedly correlated significantly with the T1 slope [9,13].

Round 2

Reviewer 1 Report

The entire paper needs to be edited.

The English language used is sometimes inappropriate (eg neonatal bed of death), and grammar, sentence structure and paragraphing is not up to publication standards.

In the introduction, the first paragraph has unreferenced statistics.

This will need to be addressed.

In the second introductory paragraph, some of the references are written while others are numbered.

 Referencing needs to be uniform throughout the paper.

In the third introductory paragraph, the first sentence is an unsupported statement of what appear to be facts. This will have to be referenced also. The purpose of the study is justified and worthy of exploration.

Author Response

Comments and Suggestions for Authors

The entire paper needs to be edited.

The English language used is sometimes inappropriate (eg neonatal bed of death), and grammar, sentence structure and paragraphing is not up to publication standards. 

Thank you for your comments

Our manuscript was rechecked by the English proofreading expert.

In the introduction, the first paragraph has unreferenced statistics.

This will need to be addressed.

I added the reference

In the second introductory paragraph, some of the references are written while others are numbered.

 Referencing needs to be uniform throughout the paper.

I corrected reference format.

In the third introductory paragraph, the first sentence is an unsupported statement of what appear to be facts. This will have to be referenced also. The purpose of the study is justified and worthy of exploration.
I added the reference.

Reviewer 2 Report

Authors have replied to most of the comments. I suggest further to make a table with all patients listed from one-X, with age, diagnosis and therapy -if any. Also add clearly how many patients underwent surgical therapy and what was the indication for X ray for outpatients. This should be clearly stated. It is also questionable what is the value of T1 slope evaluation in patients who did not went to surgery, since its assesment is primary important for cervical surgery planning.

Author Response

Comments and Suggestions for Authors

Authors have replied to most of the comments. I suggest further to make a table with all patients listed from one-X, with age, diagnosis and therapy -if any. Also add clearly how many patients underwent surgical therapy and what was the indication for X ray for outpatients. This should be clearly stated. It is also questionable what is the value of T1 slope evaluation in patients who did not went to surgery, since its assesment is primary important for cervical surgery planning.

Thank you for your comments.

This study is retrospective study. Thus, it is difficult to make a table of all patients’ data because there is not enough data available in some areas.

We added whether or not surgical treatment was finally performed, as far as we could confirm in the medical records.

Patients who present to the outpatient clinic with neck pain are eligible for this study, and the purpose of the radiography is to examine the neck pain.

As you pointed out, the most interesting point is the evaluation in surgical cases, especially how T1 slope visibility affects the changes in the postoperative period. The purpose of this study is to evaluate whether there is a baseline difference between V group and I group as a pilot evaluation before surgical treatment. Based on the results of this study, we would like to find the clinical significance of this study by conducting pre- and post-operative evaluation in surgical cases in future studies.

Reviewer 3 Report

The authors have submitted a revision of their manuscript on T1 slope seen on plain radiography.

The changes made are appreciated, but additional clarification is required for appropriate scholarly transparency. While the authors accept that their very limited retrospective study is a pilot exploratory analysis, briefly mentioning this once in the Conclusion concerns this reviewer. The nature of the study should be identified throughout the manuscript---Title, Abstract, Introduction, Methods, Results, Discussion, and Conclusion.

A more minor concern is the citation of EOS. As a software package, it should be appropriately cited in the text, such as, "analyzed using ABC Imaging Software, version 23.05.1 (Big Tech Ltd., Big City, Japan). ABC Imaging Software includes novel proprietary features that allow..."

If the software has really unique and non-standard features, the authors should additionally cite the development and validation studies that demonstrate that the features work as advertized by the company and intended by the authors of this particular manuscript.

Author Response

Comments and Suggestions for Authors

The authors have submitted a revision of their manuscript on T1 slope seen on plain radiography.

The changes made are appreciated, but additional clarification is required for appropriate scholarly transparency. While the authors accept that their very limited retrospective study is a pilot exploratory analysis, briefly mentioning this once in the Conclusion concerns this reviewer. The nature of the study should be identified throughout the manuscript---Title, Abstract, Introduction, Methods, Results, Discussion, and Conclusion.

Thank you for your comments.

We added in abstract, introduction, and discussion that this study is a pilot study.

Abstract: The T1 slope is important for cervical surgical planning, and it may be invisible on radiographic images. The prevalence of T1 invisible cases and the differences in demographic and radiographic characteristics between patients whose T1 slopes are visible or invisible remains unexplored. This pilot study aimed to evaluate the differences in these characteristics between outpatients whose T1 slopes were visible or invisible on radiographic images.

Introduciton

However, to the best of our knowledge, the percentage of patients with invisible T1 slopes and the difference in radiological characteristics between patients with visible or invisible T1 slopes remain unknown [13]. Therefore, this pilot study aimed to investigate the differences between these characteristics in outpatients with visible or invisible T1 slopes.

4. Discussion

To the best of our knowledge, this is the first study comparing the characteristics of patients with and without an identifiable T1 slope. This was a pilot study in outpatients as a preliminary step to determine the clinical significance of T1slope visibility on surgery. In this study, the T1 slope could not be identified in 46.7% of the cases. T1 slope was invisible predominantly in males. In males, the C2-7 SVA was significantly larger, and the C2-7 angle in extension was significantly smaller in cases with an invisible T1 slope. The same trend was observed in females, but the difference was insignificant. Thus, the T1 slope was invisible in the male physique, and cases with anterior cervical shift characterized by larger C2-7 SVA.

A more minor concern is the citation of EOS. As a software package, it should be appropriately cited in the text, such as, "analyzed using ABC Imaging Software, version 23.05.1 (Big Tech Ltd., Big City, Japan). ABC Imaging Software includes novel proprietary features that allow..."

If the software has really unique and non-standard features, the authors should additionally cite the development and validation studies that demonstrate that the features work as advertized by the company and intended by the authors of this particular manuscript.

We changed our manuscript as follows.

Ideally, the imaging modality enabling clear visualization of the T1 upper endplate in an upright position, such as EOS® (EOS® imaging, France), is generally desired [21]. EOS® is the EOS imaging system, a novel technique that allows for acquiring images of the body or of body sections in standing position and under normal weight-bearing conditions [22].
